# Exploring the Whole Rashomon Set of Sparse Decision Trees

**Rui Xin**[1][*]    **Chudi Zhong**[1][*]    **Zhi Chen**[1][*]

**Takuya Takagi**[2]    **Margo Seltzer**[3]    **Cynthia Rudin**[1]

[1] Duke University [2] Fujitsu Laboratories Ltd. [3] The University of British Columbia
{rui.xin926, chudi.zhong, zhi.chen1}@duke.edu
takagi.takuya@fujitsu.com, mseltzer@cs.ubc.ca, cynthia@cs.duke.edu

## Abstract

In any given machine learning problem, there might be many models that explain the data almost equally well. However, most learning algorithms return only one of these models, leaving practitioners with no practical way to explore alternative models that might have desirable properties beyond what could be expressed by a loss function. The *Rashomon set* is the set of these all almost-optimal models. Rashomon sets can be large in size and complicated in structure, particularly for highly nonlinear function classes that allow complex interaction terms, such as decision trees. We provide the first technique for completely enumerating the Rashomon set for sparse decision trees; in fact, our work provides the first complete enumeration of any Rashomon set for a non-trivial problem with a highly nonlinear discrete function class. This allows the user an unprecedented level of control over model choice among all models that are approximately equally good. We represent the Rashomon set in a specialized data structure that supports efficient querying and sampling. We show three applications of the Rashomon set: 1) it can be used to study variable importance for the set of almost-optimal trees (as opposed to a single tree), 2) the Rashomon set for accuracy enables enumeration of the Rashomon sets for balanced accuracy and F1-score, and 3) the Rashomon set for a full dataset can be used to produce Rashomon sets constructed with only subsets of the data set. Thus, we are able to examine Rashomon sets across problems with a new lens, enabling users to choose models rather than be at the mercy of an algorithm that produces only a single model.

## 1 Introduction

The *Rashomon set* is the set of almost-equally-optimal models [1, 2]. Rashomon sets were named for the *Rashomon effect* of Leo Breiman, whereby many equally good models could explain the data well [3]. The Rashomon effect helps us understand that there is not just one "best" explanation for the data, but many diverse equally predictive models. The existence of Rashomon sets has important practical implications: practitioners may not want to use the single model from the Rashomon set that was output by a machine learning algorithm. Instead, they may want to explore the Rashomon set to find models with important properties, such as interpretability, fairness, or use of specific variables, or they may want to choose a model that agrees with possible causal hypotheses, monotonicity trends, ease of calculation, or simply domain intuition. In light of viewing the learning problem this way, the whole paradigm that machine learning should provide only one "optimal" model makes little sense:

---

[*]Equal Contribution

36th Conference on Neural Information Processing Systems (NeurIPS 2022).

perhaps we should turn the *optimization* problem into a *feasibility* problem, and insist that we find all approximately-equally-good models and allow the user to choose among them. Perhaps the tiny sacrifice of a small amount of empirical risk can make the difference between a model that can be used and one that cannot.

While wishing for full Rashomon sets is easy, actually finding them can be extremely difficult for nonlinear function classes. Consider sparse decision trees, which is the focus of this work. Trees have wildly nonlinear relationships between features, where every leaf of depth $d$ represents an interaction between features of order $d$. Even for trees of depth at most 4 with only 10 binary features, the number of possible trees (the size of the hypothesis space) is more than $9.338 \times 10^{20}$ models [4]. This complexity explains why no previous work has been able to provide full Rashomon sets for optimal sparse trees, nor for any non-trivial function class with a large number of interactions.

While the hypothesis space of sparse trees is huge, the Rashomon set of good sparse trees is a small subset of it. In fact, the Rashomon set is often small enough to store, even if the function class itself is too large to be enumerated. Thus, rather than enumerate all sparse trees and store just those in the Rashomon set, we use *analytical bounds to prove that large portions of the search space do not contain any members of the Rashomon set* and can safely be excluded. This allows us to home in on just the portion of the space containing the Rashomon set. We *store the Rashomon set in a specialized data structure*, which permits memory-efficient storage and easy indexing of the Rashomon set's members. The combination of these strong bounds and efficient representation enable us to provide **the first complete enumeration of the Rashomon set for sparse decision trees on real-world datasets**. Our method is called TreeFARMS, for "Trees FAst RashoMon Sets."

The Rashomon sets we compute give us an unprecedented lens through which to examine learning problems. We can now directly answer questions such as: **Size:** How large is the Rashomon set? Does its size vary between datasets? **Variable importance:** How does variable importance change among trees within the Rashomon set? Perhaps this will give us a better sense of how important a variable is to a dataset, rather than just to one model. **Variability in predictions:** Do the models in the Rashomon set all predict similarly on the data? **Robustness:** How does the Rashomon set change if we remove a subset of data? **Rashomon sets for other losses:** How can we construct Rashomon sets for balanced accuracy and F1-score? In this work, we show how the answers to these questions can be computed directly after running our algorithm for finding the Rashomon set.

## 2   Related Work

**Rashomon sets**. Leo Breiman [3] proposed the *Rashomon effect* to describe the fact that many accurate-but-different models exist for the same data. Rashomon sets occur in healthcare, finance, criminal justice, etc. [5, 6]. They have been used for decision making [7] and robustness of estimation [8]. Semenova et al. [1] show that when the Rashomon set is large, models with various important properties such as interpretability and fairness can exist inside it. Other works use the Rashomon set to study the range of variable importance among all well-performing models [2, 9], which provides a better sense of how important a variable is in general, rather than how important it is to a specific model. Other works find diverse near-optimal solutions for integer linear programs [10, 11]. Kissel et al. [12] find collections of accurate models through model path selection. They collect models using forward selection and use these as a collection of accurate models. Our work differs from these works in that we find the *full* Rashomon set of a class of sparse decision trees on real problems.

**Model enumeration**. Ruggieri [13, 14] enumerates trees built from greedy top-down decision tree induction. Our method differs as our trees are not built using greedy induction but by searching the entire binary decision tree space, and our goal is to enumerate all well-performing trees, not simply the features in the greedy trees. Hara et al. [15, 16] enumerate linear models and rule lists or rule sets in descending order of their objective values. However, they find only one model for any given set of itemsets (association rules). Thus, they enumerate and store only a small subset of the Rashomon set.

**Decision trees**. Decision tree algorithms have a long history [17, 18, 19], but the vast majority of work on trees has used greedy induction [20, 21] to avoid solving the NP-complete problem of finding an optimal tree [22]. However, greedy tree induction provides suboptimal trees, which has propelled research since the 1990s on mathematical optimization for finding optimal decision trees [4, 23, 24, 25, 26, 27, 28, 29, 30], as well as dynamic programming with branch-and-bound [31, 32, 33, 34]. We refer readers to two recent reviews of this area [35, 36].

**Bayesian trees.** Bayesian analysis has long aimed to produce multiple almost-optimal models through sampling from the posterior distribution. While it might reasonably seem like a method such as BART [37] or random forest [38] might sample effectively from the Rashomon set, as our experiments show (Figure 1), they do not.

## 3 Bounds for Reducing the Search Space

We denote the training dataset as $\{(\mathbf{x}_i, \mathbf{y}_i)\}_{i=1}^n$, where $\mathbf{x}_i \in \{0, 1\}^p$ are binary features. Our notation uses $y_i \in \{0, 1\}$ as labels, while our method can be generalized to multiclass classification as well. Let $\ell(t, \mathbf{x}, \mathbf{y}) = \frac{1}{n}\sum_{i=1}^n \mathbf{1}[\hat{y}_i \neq y_i]$ be the loss of tree $t$ on the training set, where $\{\hat{y}_i\}_{i=1}^n$ are predicted labels given $t$. We define the objective function as the combination of the misclassification loss and a sparsity penalty on the number of leaves: $Obj(t, \mathbf{x}, \mathbf{y}) = \ell(t, \mathbf{x}, \mathbf{y}) + \lambda H_t$, where $H_t$ is the number of leaves in tree $t$ and $\lambda$ is a regularization parameter.

**Definition 1.** *(ε-Rashomon set) Let $t_{\text{ref}}$ be a benchmark model or reference model from $\mathcal{T}$, where $\mathcal{T}$ is a set of binary decision trees. The ε-Rashomon set is a set of all trees $t \in \mathcal{T}$ with $Obj(t, \mathbf{x}, \mathbf{y})$ at most $(1 + \epsilon) \times Obj(t_{\text{ref}}, \mathbf{x}, \mathbf{y})$:*

$$R_{set}(\epsilon, t_{\text{ref}}, \mathcal{T}) := \{t \in \mathcal{T} : Obj(t, \mathbf{x}, \mathbf{y}) \leq (1 + \epsilon) \times Obj(t_{\text{ref}}, \mathbf{x}, \mathbf{y})\}. \tag{1}$$

For example, if we permit models within 2% of the reference objective value, we would set $\epsilon$ to be 0.02. Note that we use $R_{\text{set}}(\epsilon)$ to represent $R_{\text{set}}(\epsilon, t_{\text{ref}}, \mathcal{T})$ when $t_{\text{ref}}$ and $\mathcal{T}$ are clearly defined. We use $\theta_\epsilon := (1 + \epsilon) \times Obj(t_{\text{ref}}, \mathbf{x}, \mathbf{y})$ to denote the threshold of the Rashomon set. Typically, the reference model is an empirical risk minimizer $t_{\text{ref}} \in \arg\min_{t \in \text{trees}} Obj(t, \mathbf{x}, \mathbf{y})$. Recent advances in decision tree optimization have allowed us to find this empirical risk minimizer, specifically using the GOSDT algorithm [31, 32]. Our goal is to store $R_{set}(\epsilon, t_{\text{ref}}, \mathcal{T})$, sample from it, and compute statistics from it.

Customized analytical bounds, leveraging tools from [31, 4], help reduce the search space for Rashomon set construction. As every possible tree is being grown in the process of dynamic programming, some of its leaves will have been determined ("fixed"), and others will not have yet been determined ("unfixed"). Bounds for these incomplete trees compare two quantities: the performance that might be achieved in the best possible case, when every unfixed part of the tree has perfect classification for all of its points, and $\theta_\epsilon$, the threshold of the Rashomon set. If the first is larger – i.e., worse – than $\theta_\epsilon$, we know that extensions of the tree will never be in the Rashomon set. Each partial tree $t$ (that is, a tree that can be extended) is represented in terms of five variables: $t_{\text{fix}}$ are the "fixed" leaves we do not extend during this part of the search (there will be a different copy of the tree elsewhere in the search where these leaves will potentially be split), $\delta_{\text{fix}}$ are the labels of the data points within the fixed leaves, $t_{\text{split}}$ are the "unfixed" leaves we could potentially split during the exploration of this part of the search space (with labels $\delta_{\text{split}}$) and $H_t$ is the number of leaves in the current tree. Thus, our current position in the search space is a partial tree $t = (t_{\text{fix}}, \delta_{\text{fix}}, t_{\text{split}}, \delta_{\text{split}}, H_t)$. The theorems below allow us to exclude large portions of the search space.

**Theorem 3.1.** *(Basic Rashomon Lower Bound) Let $\theta_\epsilon$ be the threshold of the Rashomon set. Given a tree $t = (t_{\text{fix}}, \delta_{\text{fix}}, t_{\text{split}}, \delta_{\text{split}}, H_t)$, let $b(t_{\text{fix}}, \mathbf{x}, \mathbf{y}) := \ell(t_{\text{fix}}, \mathbf{x}, \mathbf{y}) + \lambda H_t$ denote the lower bound of the objective for tree $t$. If $b(t_{\text{fix}}, \mathbf{x}, \mathbf{y}) > \theta_\epsilon$, then the tree $t$ and all of its children are not in the ε-Rashomon set.*

We can tighten the basic Rashomon lower bound by using knowledge of *equivalent points* [39]. Data points are equivalent if they have exactly the same feature values. Let $\Omega$ be a set of leaves. *Capture* is an indicator function that equals 1 if $\mathbf{x}_i$ falls into one of the leaves in $\Omega$, and 0 otherwise, in which case we say that $\text{cap}(\mathbf{x}_i, \Omega) = 1$. Let $e_u$ be a set of equivalent points and $q_u$ be the minority class label among points in $e_u$. A dataset consists of multiple sets of equivalent points. Let $\{e_u\}_{u=1}^U$ enumerate these sets. The bound below incorporates equivalent points.

**Theorem 3.2.** *(Rashomon Equivalent Points Bound) Let $\theta_\epsilon$ be the threshold of the Rashomon set. Let $t$ be a tree with leaves $t_{\text{fix}}, t_{\text{split}}$ and lower bound $b(t_{\text{fix}}, \mathbf{x}, \mathbf{y})$. Let $b_{equiv}(t_{\text{split}}, \mathbf{x}, \mathbf{y}) := \frac{1}{n}\sum_{i=1}^n \sum_{u=1}^U cap(\mathbf{x}_i, t_{\text{split}}) \wedge \mathbb{1}[\mathbf{x}_i \in e_u] \wedge \mathbb{1}[y_i = q_u]$ be the lower bound on the misclassification loss of the unfixed leaves. Let $B(t, \mathbf{x}, \mathbf{y}) := b(t_{\text{fix}}, \mathbf{x}, \mathbf{y}) + b_{equiv}(t_{\text{split}}, \mathbf{x}, \mathbf{y})$ be the Rashomon lower bound of $t$. If $B(t, \mathbf{x}, \mathbf{y}) > \theta_\epsilon$, tree $t$ and all its children are not in the ε-Rashomon set.*

We can use this bound recursively on all subtrees we discover during the process of dynamic programming. If, at any time, we find the sum of the lower bounds of subproblems created by a split

exceed the threshold of the Rashomon set, the split that led to these subproblems will never produce any member of the Rashomon set. This is formalized in the Rashomon Equivalent Points Bound for Subtrees, Theorem E.1 in Appendix E, which dramatically helps reduce the search space. We also use the "lookahead" bound, used in GOSDT [31], which looks one split forward and cuts all the children of tree $t$ if $B(t, \mathbf{x}, \mathbf{y}) + \lambda > \theta_\epsilon$.

## 4 Storing, Extracting, and Sampling the Rashomon Set

The key to TreeFARMS's scalability is a novel *Model Set* representation. The **Model Set (MS)** is a set of hierarchical maps; each map is a **Model Set Instance (MSI)**. Conceptually, we identify a MSI by a <subproblem, objective> pair; in reality, we use pointers to improve execution time and reduce memory consumption. A MSI can represent a terminal (leaf) node, an internal node, or both (See Appendix A). A leaf MSI stores only the subproblem's prediction and the number of false positives and negatives (or false predictions of each class in the case of multiclass classification). An internal MSI, $M$, is a map whose keys are the features on which to split the subproblem and whose values are an array of pairs, each referring to left and right MSIs whose objectives sum to the objective of $M$. TreeFARMS's efficiency stems from the fact that the loss function for decision trees takes on a discrete number of values (approximately equal to the number of samples in the training data set), while the number of trees in the Rashomon set is frequently orders of magnitude larger. Therefore, many trees (and subtrees) have the same objective. By grouping together trees with the same objective, we avoid massive amounts of data duplication and computation. See Appendix A for an example. Equipped with these data structures, we now present our main algorithm.

### 4.1 TreeFARMS Implementation

We implement TreeFARMS in GOSDT [31], which uses a dynamic-programming-with-bounds formulation to find the optimal sparse decision trees. Each *subproblem* is defined by a support set $s_a \in \{0, 1\}^n$ such that the $i^{th}$ element is 1 iff $\mathbf{x}_i$ satisfies the Boolean assertion $a$ that corresponds to a decision path in the tree. For each subproblem in the dynamic program, GOSDT keeps track of upper and lower bounds on its objective. It stores these subproblems and their bounds in a *dependency graph*, which expresses the relationships between subproblems. TreeFARMS transforms GOSDT in two key ways. First, while searching the space, TreeFARMS prunes the search space by removing only those subproblems whose objective lower bound is greater than the thresholds defined by the Rashomon set bound, $\theta_\epsilon$, rather than GOSDT's objective-based upper bound. Second, rather than finding the *single* best model expressed by the dependency graph, TreeFARMS returns *all models in the Rashomon set defined by $\theta_\epsilon$*.

### 4.2 Extraction Algorithm

TreeFARMS (Alg. 1) constructs the dependency graph using the bounds from Section 3, and Extract (Alg. 2) extracts the Rashomon set from the dependency graph.

**TreeFARMS (Algorithm 1)**: **Line 1**: Call GOSDT to find the best objective. **Line 2**: Using the best objective from Line 1, compute $\theta_\epsilon$, as defined in Definition 1. **Lines 3-6**: Configure and execute (the modified) GOSDT to produce a dependency graph containing all subproblems in the Rashomon set. **Lines 7-10**: Initialize the parameters needed by `extract` and then call it to extract the Rashomon set from the dependency graph.

**Extract (Algorithm 2)**: We present an abbreviated version of the algorithm here with the full details in Appendix B. **Line 1**: Check to see if we already have the Rashomon set for the given problem and scope; if so, return immediately. **Lines 2-3**: If we can make a prediction for the given subproblem that produces loss less than or equal to scope (using Theorems 3.1 and 3.2), then the leaf for this subproblem should be part of trees in the Rashomon set, so we add it to our Model Set. **Lines 4-12**: Loop over each feature and consider splitting the current problem on that feature. **Lines 5-6**: Skip over any splits that either do not appear in the dependency graph or whose objectives produce a value greater than scope (using Theorem E.1). **Lines 7-9**: Find all subtrees for left and right that should appear in trees in the Rashomon set, and construct the set of MSI identifiers for each. **Lines 10-12**: Now, take the cross product of the sets of MSI identifiers. For each pair, determine if the sum of the objectives for those MSI are within scope (using Theorem E.1). If so, we add the left/right pair to

the appropriate MSI, creating a new MSI if necessary. When this loop terminates, all trees in the Rashomon set are represented in MS.

## 4.3 Sampling from the Rashomon Set

If we can store the entire Rashomon set in memory, then sampling is unnecessary. However, sometimes the set is too large to fit in memory (e.g., the COMPAS data set [40] with a regularization of 0.005 and a Rashomon threshold that is within 15% of optimal produces $10^{12}$ trees). Our Model Set representation permits easy uniform sampling of the Rashomon set that can be used to explore the set with a much lower computational burden. Appendix C presents a sampling algorithm.

---

**Algorithm 1** TreeFARMS$(\mathbf{x}, \mathbf{y}, \lambda, \epsilon) \rightarrow R_{\text{set}}$

---

*// Given a dataset $(\mathbf{x}, \mathbf{y})$, $\lambda$, and $\epsilon$, return the set, $R_{set}$, of all trees whose objective is in $\theta_\epsilon$.*
1: $opt \leftarrow gosdt.optimal\_obj(\mathbf{x}, \mathbf{y}, \lambda)$ *// Use GOSDT to find opt, the objective of the optimal tree.*
2: $\theta_\epsilon \leftarrow opt * (1 + \epsilon)$ *// Compute $\theta_\epsilon$, which is the threshold of the Rashomon Set.*
3: $gosdt.best\_current\_obj \leftarrow \theta_\epsilon$ *// Set and fix GOSDT's best current objective $R^c$ to $\theta_\epsilon$*
*// Disable leaf accuracy bound and incremental accuracy bound in GOSDT (see Algorithm 7 in [31]), which are used to find optimal trees but could remove near-optimal trees in the Rashomon set*
4: $gosdt.leaf\_accuracy \leftarrow False$ *// Disable leaf accuracy bound.*
5: $gosdt.incremental\_accuracy \leftarrow False$ *// Disable incremental accuracy bound.*
6: $gosdt.fit(\mathbf{x}, \mathbf{y}, \lambda)$ *// Run the branch-and-bound algorithm using new settings of bounds.*
7: $G \leftarrow gosdt.get\_graph()$ *// Return dependency graph obtained from branch-and-bound.*
8: global $MS \leftarrow \{\}$ *// Initialize Rashomon Model Set $MS$, which is a global data structure.*
9: $P \leftarrow ones(|\mathbf{y}|)$ *// Create the subproblem representation for the entire dataset.*
10: extract$(G, P, \theta_\epsilon)$ *// Fill in $MS$ with trees in the Rashomon set.*
11: **return** $MS_P$

---

---

**Algorithm 2** extract$(G, sub, scope)$ (Detailed algorithm in Appendix B)

---

*// Given a dependency graph, $G$; a subproblem, $sub$; and a maximum allowed objective value, $scope$, populate the global variable $MS$ with the Rashomon set for sub within scope.*
*// Check if we have already solved the subproblem $sub$. SOLVED is presented in Alg. 3.*
1: **if** SOLVED$(MS, sub, scope)$ **then return**
2: **if** $G[sub] \leq scope$ **then** *// Check if we should create a leaf. (Theorems 3.1 and 3.2).*
3:   $MS \leftarrow MS \cup newLeaf(sub)$ *// newLeaf is presented in Alg. 3.*
*// Consider splits on each feasible split feature skipping those not in G or with bounds too large.*
4: **for** each feature $j \in [1, M]$ **do**
5:   $sub_l, sub_r \leftarrow split(sub, j)$
6:   **if** either $sub_l, sub_r$ **not in** $G$ **or** $G[sub_l] + G[sub_r] > scope$ **then continue**
    *// Find Model Sets Instances for left and right.*
7:   $extract(G, sub_l, scope - G[sub_r])$
8:   $extract(G, sub_r, scope - G[sub_l])$
9:   $left, right \leftarrow MS_{sub_l}, MS_{sub_r}$
10:   **for** each $(m_l, m_r) \in (left \times right)$ **do** *// Consider cross product of left/right MSI.*
    *// Skip trees with objective outside of scope. (Theorem E.1).*
11:     **if** $obj(m_l) + obj(m_r) > scope$ **then continue**
12:     $MS \leftarrow MS \cup add(sub, obj(m_l) + obj(m_r), m_l, m_r)$ *// Add pair to Model Set.*
*// $MS$ now contains all MSI for sub with objective less than or equal to scope.*
13: **return**

---

# 5 Applications of the Rashomon Set

Besides allowing users an unprecedented level of control over model choice, having access to the Rashomon set unlocks powerful new capabilities. We present three example applications here.

## 5.1 Variable Importance for Models in the Rashomon Set via Model Class Reliance

The problem with classical variable importance techniques is that they generally provide the importance of one variable to one model. However, just because a variable is important to one model does not mean that it is important in general. To answer this more general question, we consider model class reliance (MCR) [2]. MCR provides the range of variable importance values across the set of all well-performing models. $MCR_-$ and $MCR_+$ denote the lower and upper bounds of this range, respectively. A feature with a large $MCR_-$ is important in all well-performing models; a feature with a small $MCR_+$ is unimportant to every well-performing model. Past work managed to calculate $MCR_-$ only for convex loss in linear models [2]. For decision trees, the problem is nonconvex and intractable using previous methods. A recent work [41] estimates MCR for random forest, leveraging the fact that the forest's trees are grown greedily from the top down; the optimized sparse trees we consider are entirely different, and as we will show, many good sparse trees are difficult to obtain through sampling. Since TreeFARMS can enumerate the whole Rashomon set of decision trees, we can *directly* calculate variable importance for every tree in the Rashomon set and then find its minimum and maximum to compute the MCR, as described in Appendix D. See Section 6.2 for the results. If the Rashomon set is too large to enumerate, sampling (Section 4.3) can be used to obtain sample estimates for the MCR (shown in Section 6.2).

## 5.2 Rashomon Sets Beyond Accuracy: Constructing the Rashomon Set for Other Metrics

For imbalanced datasets, high accuracy is not always meaningful. Metrics such as balanced accuracy and F1-score are better suited for these datasets. We show that, given the Rashomon set constructed using accuracy, we can directly find the Rashomon sets for balanced accuracy and F1-score.

Let $q^+$ be the proportion of positive samples and $q^-$ be the proportion of negative samples, i.e., $q^+ + q^- = 1$. Let $q_{\min} := \min(q^+, q^-)$ and $q_{\max} := \max(q^+, q^-)$. Let $FPR$ and $FNR$ be the false positive and false negative rates. We define the Accuracy Rashomon set as $A_\theta := \{t \in \mathcal{T} : q^- FPR_t + q^+ FNR_t + \lambda H_t \leq \theta\}$, where $\theta$ is the objective threshold of the Accuracy Rashomon set, similar to $\theta_\epsilon$ in Section 3. The next two theorems guide us to find the Balanced Accuracy or F1-Score Rashomon set from the Accuracy Rashomon set. Their proofs are presented in Appendix E. We use $\delta$ in these theorems to denote the objective threshold of Balanced Accuracy or F1-Score Rashomon sets.

**Theorem 5.1.** *(Accuracy Rashomon set covers Balanced Accuracy Rashomon set) Let* $B_\delta := \{t \in \mathcal{T} : \frac{FPR_t + FNR_t}{2} + \lambda H_t \leq \delta\}$ *be the Balanced Accuracy Rashomon set. If*

$$\theta \geq \min\left(2q_{\max}\delta, q_{\max} + (2\delta - 1)q_{\min} + (1 - 2q_{\min})\lambda 2^d\right),$$

*where $d$ is the depth limit, then* $\forall t \in B_\delta, t \in A_\theta$.

**Theorem 5.2.** *(Accuracy Rashomon set covers F1-Score Rashomon set) Let*

$$F_\delta := \left\{t \in \mathcal{T} : \frac{q^- FPR_t + q^+ FNR_t}{2q^+ + q^- FPR_t - q^+ FNR_t} + \lambda H_t \leq \delta\right\}$$

*be the F1-score Rashomon set. Suppose $q^+ \in (0, 1)$, $q^- \in (0, 1)$, and $\delta - \lambda H_t \in (0, 1)$. If* $\theta \geq \min\left(\max\left(\frac{2q^+\delta}{1-\delta}, \frac{2q^+(\delta - \lambda 2^d)}{1-(\delta-\lambda 2^d)}\right) + \lambda 2^d\right), \mathbb{1}[\delta < \sqrt{2} - 1]\frac{2\delta}{1+\delta} + \mathbb{1}[\delta \geq \sqrt{2} - 1](\delta + 3 - 2\sqrt{2})\right)$, *then* $\forall t \in F_\delta, t \in A_\theta$.

We can use Theorems 5.1 and 5.2 above to find all trees in the Balanced Accuracy or F1-score Rashomon set directly by searching through the Accuracy Rashomon set with objective threshold $\theta$ that satisfies the inequality constraint. In our implementation, we first use GOSDT to find the optimal tree w.r.t. misclassification loss and then calculate its objective w.r.t. balanced accuracy and F1-score. Then we set $\delta$ and the corresponding $\theta$. This guarantees that the Rashomon sets for balanced accuracy and F1-score objective are not empty. Experiments in Section 6.3 illustrate this calculation.

## 5.3 Sensitivity to Missing Groups of Samples

Though sparse decision trees are usually robust (since predictions are made separately in each leaf), we are also interested in how a sample or a group of samples influences all well-performing models

(i.e., whether this subset of points is *influential* [42].) Influence functions cannot be calculated for decision trees since they require differentiability. The following two theorems help us find optimal or near-optimal trees for a dataset in which a group of instances has been removed by searching through the Rashomon set obtained from the full dataset. Here, we consider the misclassification loss.

Let $\tilde{t}^*$ be the optimal tree trained on the reduced dataset $\{\mathbf{x}_{[\backslash K,\cdot]}, \mathbf{y}_{[\backslash K]}\}$ where $K$ is a set of indices of instances that we remove from the original dataset. Let $|K|$ denote the cardinality of the set $K$. Overloading notation to include the dataset, let $R_{set}(\epsilon, t^*, \mathcal{T}, \mathbf{x}, \mathbf{y}) = R_{set}(\epsilon, t^*, \mathcal{T})$ (see Eq (1)) be the Rashomon set of the original dataset, where $t^*$ is the optimal tree trained on the original dataset. We define the $\epsilon'$-Rashomon set on the reduced dataset as

$$R_{set}(\epsilon', \tilde{t}^*, \mathcal{T}, \mathbf{x}_{[\backslash K,\cdot]}, \mathbf{y}_{[\backslash K]}) := \left\{ t \in \mathcal{T} : Obj(t, \mathbf{x}_{[\backslash K,\cdot]}, \mathbf{y}_{[\backslash K]}) \leq (1+\epsilon') \times Obj(\tilde{t}^*, \mathbf{x}_{[\backslash K,\cdot]}, \mathbf{y}_{[\backslash K]}) \right\}.$$

**Theorem 5.3.** *(Optimal tree after removing a group of instances is still in full-dataset Rashomon set) If $\epsilon \geq \frac{2|K|}{n \times Obj(t^*, \mathbf{x}, \mathbf{y})}$, then $\tilde{t}^* \in R_{set}(\epsilon, t^*, \mathcal{T}, \mathbf{x}, \mathbf{y})$.*

Now we consider not only the optimal tree on the reduced dataset but also the near-optimal trees.

**Theorem 5.4.** *(Rashomon set after removing a group of instances is within full-dataset Rashomon set) If $\epsilon \geq \epsilon' + \frac{(2+\epsilon')|K|}{n \times Obj(t^*, \mathbf{x}, \mathbf{y})}$, then $\forall t \in R_{set}(\epsilon', \tilde{t}^*, \mathcal{T}, \mathbf{x}_{[\backslash K,\cdot]}, \mathbf{y}_{[\backslash K]}), t \in R_{set}(\epsilon, t^*, \mathcal{T}, \mathbf{x}, \mathbf{y})$.*

# 6 Experiments

Our evaluation answers the following questions: 1. How does TreeFARMS compare to baseline methods for searching the hypothesis space? (§6.1), 2. How quickly can we find the entire Rashomon set? (§6.1), 3. What does the Rashomon set look like? What can we learn about its structure? (§G.2), 4. What does MCR look like for real datasets? (§6.2), 5. How do balanced accuracy and F1-score Rashomon sets compare to the accuracy Rashomon set? (§6.3), and 6. How does removing samples affect the Rashomon set? (§6.4).

Finding the Rashomon set is computationally difficult due to searching an exponentially growing search space. We use datasets from the UCI Machine Learning Repository [Car Evaluation, Congressional Voting Records, Monk2, and Iris, see 43], a penguin dataset [44], a criminal recidivism dataset [COMPAS, shared by 40], the Fair Isaac (FICO) credit risk dataset [45] used for the Explainable ML Challenge, and four coupon datasets (Bar, Coffee House, Cheap Restaurant, and Expensive Restaurant) [46] that come from surveys. More details are in Appendix F.

## 6.1 Performance and Timing Experiments

Our method directly constructs the Rashomon set of decision trees of a dataset. While, to the best of our knowledge, there is no previous directly comparable work, there are several methods one might naturally consider to find this set. One might use methods that sample from the high-posterior region of tree models, though we would not know how many samples we need to extract the full Rashomon set. Thus, the first baseline method we consider is sampling trees from the posterior distribution of Bayesian Additive trees [47, 48]. We used the R package BART [37], setting the number of trees in each iteration to 1. Many ensemble methods combine a diverse set of trees. The diversity in this set comes from fitting on different subsets of data. Trees produced by these methods would be natural approaches for finding the Rashomon set. We thus generated trees from three different methods (Random Forest [38], CART [20], and GOSDT [31]), on many subsets of our original data.

Figure 1 compares the Rashomon set with the four baselines on the Monk2, COMPAS, and Bar datasets. We show the number of distinct trees versus the objective value. We sort the trees with respect to their objective values, so all methods show an increasing trend. TreeFARMS (in purple) found ***orders of magnitude more distinct trees in the Rashomon set*** than any of the four baselines on all of the datasets. The baseline methods tend to find many duplicated trees. For example, in 46 seconds, BART finds only 488 distinct trees on Monk2, whereas TreeFARMS found $10^8$. Other methods find ∼20,000 distinct trees. Most trees found by the baselines are not even in the Rashomon set, i.e., most of their trees have objective values higher than the threshold of the Rashomon set.

Figure 2 shows run times, specifically, Fig. 2(c) shows that TreeFARMS ***finds trees in the Rashomon set at a dramatically faster rate*** than the baselines. Appendix G.1 has more results.

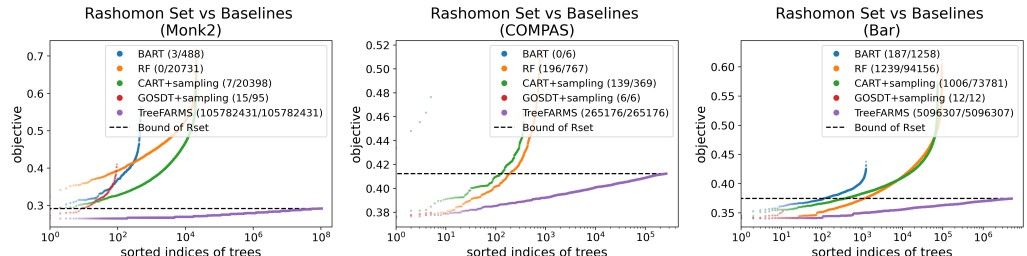

Figure 1: Comparison of trees in the Rashomon set ($\lambda = 0.01, \epsilon = 0.1$) and trees generated by baselines. Trees in the Rashomon set have objective below the dashed line. (A/B) in legend represents that A trees among B trees trained by the baseline are in the Rashomon set. For example, RF (196/767) means 196 trees among 767 distinct trees trained by Random Forest are in the Rashomon set. Indices are in log scale to accommodate differences in orders of magnitude of tree counts among methods.

The takeaway from this experiment is that ***natural baselines find at best a tiny sliver of the Rashomon set***. Further, ***the way we discovered this was to develop a method that actually enumerates the Rashomon set***. We would not have known, using any other way we could think of, that sampling-based approaches barely scratch the surface of the Rashomon set.

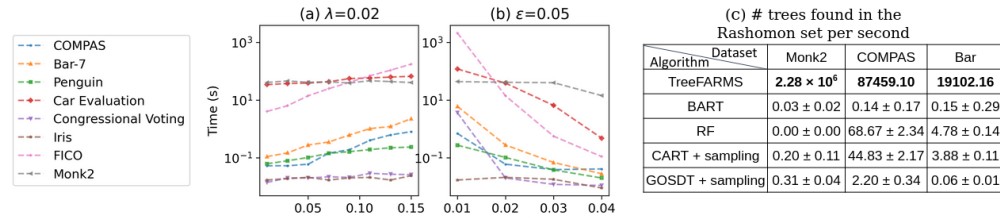

Figure 2: (a), (b) Run times for computing Rashomon sets as a function of $\epsilon$ and $\lambda$ respectively. (c) Number of trees in the Rashomon set found by each method per second. The total time is approximately 46, 3, and 270 seconds for Monk2, COMPAS, and Bar respectively. TreeFARMS is the only algorithm *guaranteed* to find all trees in the Rashomon set. (Appendix G.1 has results for all the datasets.)

## 6.2 Variable Importance: Model Class Reliance

Without TreeFARMS, it has not been possible to compute overall variable importance calculations such as MCR for complex function classes with interaction terms such as decision trees. Here, we exactly compute MCR on the COMPAS and Bar datasets (see Figure 3). For the COMPAS dataset (left subfigure)features related to prior counts generally have high $MCR_+$, which means these features are very important for some trees in the Rashomon set. For the Bar dataset (right subfigure), features "Bar_1-3" and "Bar_4-8" have dominant $MCR_+$ and $MCR_-$ compared with other features, indicating that for all well-performing trees, these features are the most important. This makes sense, since people who go to bars regularly would be likely to accept a coupon for a bar.

**Sampling for MCR**: Sampling has a massive memory benefit over storing the whole Rashomon set, because we do not need to store the samples. Since MCR requires computing extreme value statistics (max and min over the Rashomon set), it poses a test for the sampling technique posed in Section 4.3. Figure 3 shows sampled MCR and its convergence to true MCR. 25% of the samples are usually sufficient. More results are shown in Appendix G.3.

## 6.3 Balanced Accuracy and F1-score Rashomon set from Accuracy Rashomon set

As discussed in Section 5.2, Rashomon sets of balanced accuracy and F1-Score are contained in the Rashomon set for accuracy. Figure 4 shows trees in the Accuracy Rashomon set, which covers the Balanced Accuracy Rashomon set (left) and F1-score Rashomon set (right). The black dashed line indicates the corresponding objective thresholds and blue dots below the dashed line are trees

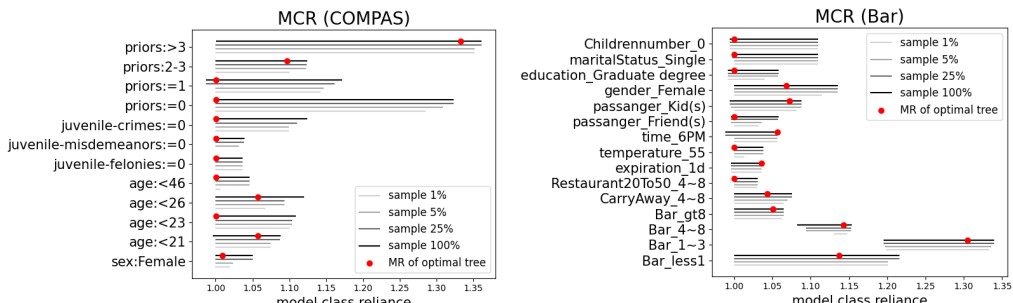

Figure 3: Variable Importance: Model class reliance on the COMPAS and Bar ($\lambda = 0.01, \epsilon = 0.05$). Red dots indicate the model reliance (variable importance) calculated from the optimal tree. Each line connects MCR$_-$ and MCR$_+$ showing the range of variable importance among all good models.

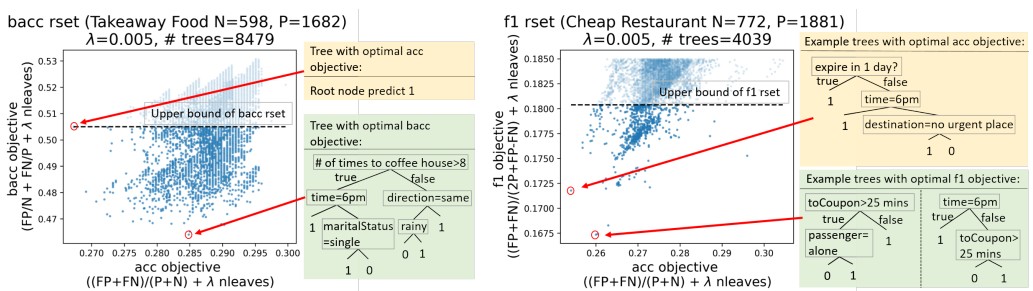

Figure 4: Example of Balanced Accuracy Rashomon set (left) and F1-Score Rashomon set (right). # trees indicates the number of trees in the Balanced Accuracy or F1-score Rashomon set. Trees in the yellow region have optimal accuracy objective and trees in the green region have optimal balanced accuracy or F1-score objective.

in these Rashomon sets. Note that the tree with the minimum misclassification objective is not the tree that optimizes other evaluation metrics. For example, in the left subfigure, a single root node that predicts all samples as 1 has the optimal accuracy objective (in yellow), while another 6-leaf tree minimizes the balanced accuracy objective (in green). Actually, many trees have better balanced accuracy objective than the tree that minimizes the accuracy objective. A similar pattern holds for the F1-score Rashomon set (see right subfigure). Some trees that have worse accuracy objectives are better in terms of the F1-score objective. More figures are shown in Appendix G.4.

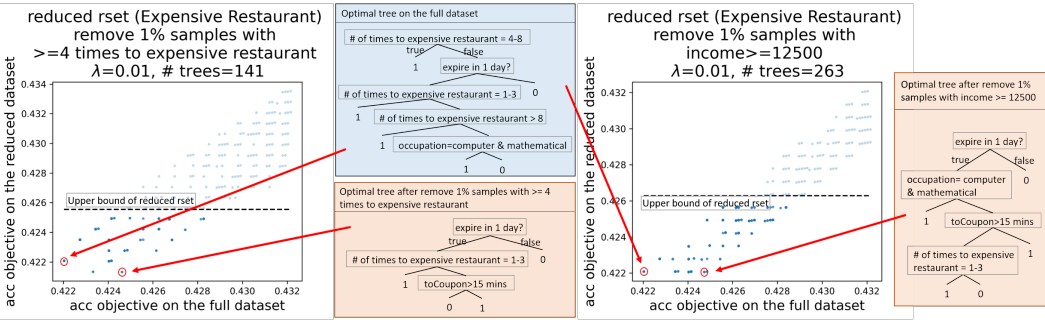

Figure 5: Example Rashomon sets and optimal trees after we remove the 1% of samples with "number of times to expensive restaurant $\geq 4$" (left) and "income $\geq \$12,500$" (right) on the Expensive Restaurant dataset. The optimal tree on the full dataset is shown in the gray region and optimal trees on the corresponding reduced datasets are in the orange region.

### 6.4 Rashomon set after removing a group of samples

Figure 5 shows accuracy objective on the full dataset versus the objective on the reduced dataset after 1% of samples with "number of times to expensive restaurant $\geq 4$" (left) and "income $\geq$ \$12,500" (right) are removed. The black dashed line indicates the objective threshold of the reduced Rashomon set and blue dots below the dashed line are trees in the reduced Rashomon set. As we can see, both scatter plots show a high correlation between the accuracy objective on the full dataset and the reduced dataset, indicating sparse near-optimal trees are robust to the shift in sample distribution. In other words, well-performing trees trained on the full dataset are usually still well-performing if some samples are removed. Optimal trees on the reduced dataset might be different, as we see by comparing the trees in the orange region and blue region. More results are shown in Appendix G.5.

## 7   Conclusion

This work opens the door to interesting discussions on variable importance, distributional shift, and user options. By efficiently representing *all* optimal and slightly suboptimal models for complex nonlinear function classes with interactions between variables, we provide a range of new user-centered capabilities for machine learning systems, and a new understanding of the importance of variables. Importantly, TreeFARMS allows users a *choice* rather than handing them a single model.

## Acknowledgements

We thank the anonymous reviewers for their suggestions and insightful questions. The authors acknowledge funding from the National Science Foundation under grants IIS-2147061 and IIS-2130250, National Institute on Drug Abuse under grant R01 DA054994, Department of Energy under grants DE-SC0021358 and DE-SC0023194, and National Research Traineeship Program under NSF grants DGE-2022040 and CCF-1934964. We acknowledge the support of the Natural Sciences and Engineering Research Council of Canada (NSERC). Nous remercions le Conseil de recherches en sciences naturelles et en génie du Canada (CRSNG) de son soutien.

## Code Availability

Implementations of TreeFARMS is available at https://github.com/ubc-systopia/treeFarms.

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
