# OpenReview forum: "Exploring the Whole Rashomon Set of Sparse Decision Trees"
_NeurIPS.cc/2022/Conference — NeurIPS 2022 Accept_

### Official Review · Reviewer_PEzG · 2022-07-06

**Rating:** 6
**Confidence:** 3
**Soundness:** 4 excellent
**Presentation:** 3 good
**Contribution:** 3 good

**Summary:**

The authors propose the TreeFARMS method, which is able to enumerate the Rashomon set of all sparse decision trees over binary variables and labels that are optimal within a user-provided multiplicative factor in terms of 0-1 loss, which they also show to be sufficient for the derivation of balanced accuracy and F1 score. This algorithm employs a branch-and-bound search over the search state of trees represented as their leaves, some of which are fixed and some as of yet "unfixed"; it then uses a simple bound (that ignores "unfixed" leaves) and another that tightens the former by considering data point groups with identical features (termed equivalent points). TreeFARMS heavily relies on implementation details of the existing GOSDT algorithm and extensions for storing different decision trees. The authors show that their proposal outperforms heuristic methods in terms of accuracy, and demonstrate the model class reliance of the variables on an intuitive example dataset.

**Questions:**

1. What is the sampling distribution? If not uniform among the entire Rashomon set, what is your intuition on which models are preferably included?

2. What is the relation between your method and the MCR method of Smith et al. for Random Forests?

[1] Gavin Smith, Roberto Mansilla, and James Goulding. Model Class Reliance for Random Forests.


**Limitations:**

The manuscript does not include a discussion of the limitations of the proposal.

**Strengths And Weaknesses:**

First work on providing the complete Rashomon set for optimal sparse trees.

Provide theoretical foundations for deriving the Rashomon set for other losses (balanced accuracy+F1).

Provide experimental evaluations for theoretically interesting applications of the Rashomon set: variable importance, and sensitivity to missing values.

Although overall solid work, the key novelty of the manuscript is limited to Sections 5.2 and 5.3 as it otherwise effectively extends an existing algorithm to enumerate the Rashomon set and the investigation of existing MCR.

The writing is good in general, but the authors are guilty of overselling; the manuscript will improve by removing exaggerations (e.g., “at the mercy of an algorithm”, “only scratch the surface of”) or allegations “this complexity explains why no previous works”, in lieu of a formal complexity result.

A clear limitation is the use of only binary labels and variables. The experiments show that setting the multiplicative and regularisation parameters provides little a-priori control on the number of models in the resulting set (e.g. it can be up to 10e12 in given configurations, line 190).

The statistics during sampling are not clear to this reviewer.

---

> ### Author Response · Authors · 2022-08-02
> **Response to reviewer PEzG**
>
> Thank you so much for your review! We really appreciate it! See below for our response to your questions.
>
> > Q: Although overall solid work, the key novelty of the manuscript is limited to Sections 5.2 and 5.3 as it otherwise effectively extends an existing algorithm to enumerate the Rashomon set and the investigation of existing MCR.
>
> *We presented the first algorithm to enumerate the Rashomon set of sparse decision trees*. If you know of another algorithm that stores the full set of models from *any* significant model class (including trees), please tell us -- we do not know of any. Our algorithm does not extend GOSDT per se; it uses an extended version of a data structure from GOSDT to do something for which GOSDT was never designed.
> In particular, GOSDT *eliminates almost all of* the search space (to produce one tree), while TreeFARMS *finds and stores* the whole Rashomon set.
> To achieve this goal, we developed a new data structure to store and extract trees. Without the Model Set representation, we can not enumerate large Rashomon sets such as the $10^9$ trees for monk2 with $\lambda=0.01$ as shown in Figure 10.
>
> > Q: A clear limitation is the use of only binary labels and variables.
>
> The labels need not be binary. The algorithm can be used for multi-class labels; **iris** and **penguin** datasets used in our experiments have three classes. In the main paper, we define the label to be binary for ease of exposition. We can definitely make this point clearer in the main paper. Also, for real-valued variables, we split them by using the midpoints of every two consecutive values; for categorical variables, we transform them to one-hot encoded binary variables. This is standard practice in decision tree optimization (see Verwer and Zhang, 2019 and Aglin, Nijssen and Schaus, 2020.)
>
> > The experiments show that setting the multiplicative and regularisation parameters provides little a-priori control on the number of models in the resulting set (e.g., it can be up to 10e12 in given configurations, line 190).
>
> It's impossible to know the exact size of the Rashomon set before actually calculating it. (How many close-to-optimal models are there? It's hard to answer beforehand.) The results in Figure 10 show how the size of the Rashomon set changes w.r.t multiplier and regularization. In general, we expect that the size of the Rashomon set increases exponentially as the multiplier increases and as regularization decreases.
>
> > Q: The statistics during sampling are not clear to this reviewer. What is the sampling distribution? If not uniform among the entire Rashomon set, what is your intuition on which models are preferably included?
>
> We uniformly sampled the Rashomon Set to get a good representation of the set. See line 191 in the main paper. One advantage of TreeFARMS is that we can sample uniformly in the Rashomon Set without enumerating all trees.
>
> > Q: What is the relation between your method and the MCR method of Smith et al$.$ for Random Forests?
>
> Smith et al$.$ *estimate* MCR for a whole forest. We *calculate* MCR for the set of sparse trees. Once the Rashomon set is constructed, we can directly calculate the exact MCR. There is no estimation or approximation needed.
> The method of Smith et al. cannot handle single sparse trees, since random forests build trees greedily from the top down rather than optimizing them like we do. We will cite this paper though, it's a great paper.
>
> Thank you so much once again for your review!

---

### Official Review · Reviewer_4Gdc · 2022-07-11

**Rating:** 9
**Confidence:** 5
**Soundness:** 4 excellent
**Presentation:** 3 good
**Contribution:** 4 excellent

**Summary:**

This paper presents the first algorithm TreeFARMS, a branch & bound algorithm with memoization (or dynamic programming with pruning bounds) to completely enumerate the Rashomon set of sparse decision trees. The Rashomon set is the set of almost-equally as good trees as the given reference tree. As recently proposed, the GOSDT algorithm [3][32] can find the empirical risk minimizer among sparse decision trees. TreeFARMS use this optimal tree as the reference for the Rashomon set, and also use its dependency graph as a model search space. The basic idea is to decompose search for decision trees into subproblems, and recursively solve child problems with maintaining these problems (dependency graph) and with pruning problem space with bounds. Extending this search strategy with a specialized data structure called Model Set (MS) representation enables the full enumeration of the Rashomon set of all sparse decision trees in the search space of GOSDT. Clear use cases of this complete enumeration such as the size of the Rashomon set, the variability of feature importances, the variability of predictions, and the sample dependency of the Rashomon set are further provided and discussed in detail.

**Questions:**

- Because the title includes "sparse", and thus could you explain why enumeration is limited to "sparse", and what does the paper mean by the word "sparse", or explicitly (but concisely?) explain the dependency inherited from GOSDT?

- Could you add the table of contents for the supplementary document? I wanted to know the results for "the size of Rashomon set", which is the first example use-case mentioned in Introduction, but at the first glance, they are sort of hidden by other detailed information.

- Is the "dependency graph" actually a tree, a DAG, or a cyclic graph? I'm not sure about the connection, but I felt some abstract similarity to topics such as decision diagram (DD) representation of logical functions and traversing an enumeration tree, the search space of gSpan algorithm for all subgraph patterns, with bounds as in https://doi.org/10.1109/TPAMI.2016.2567399 for example.



**Limitations:**

No specific concerns on limitations and negative societal impact of this work.


**Strengths And Weaknesses:**

### Strengths

- This paper presents the first very powerful algorithm to completely enumerate the Rashomon set of optimal sparse decision trees thanks to GOSDT algorithm, pruning bounds, and MS data structures. This would be quite impactful because the instability or multiplicity of good models, for decision trees in particular, is a long standing interests in decision tree learning since Breiman.

- The paper also showed several clear use cases with detailed analysis. Full enumeration of the Rashomon set enables to analyze the size of Rashomon set, the variability of feature importances, the variability of predictions, and the sensitivity to missing groups of samples.


### Weaknesses

- Though the paper comes with a detailed supplementary document, some descriptions are hard to follow at the first glance due to the strong dependency on GOSDT. For example, the main algorithm of TreeFARMS, Algorithm 1, includes undefined procedure such as gosdt.get_graph or gosdt.configure_bounds. Since this is the main result, much more self-contained explanations would be appreciated.

- Also due to the implicit dependency on GOSDT, several important points would be unclearly presented. For example, one big issue is why the paper title includes "sparse" although Algorithm 1 and 2 have no sparsity-inducing parts apparently. If I understand it, the model search space for complete enumeration comes from GOSDT targeting sparse decision trees, and thus, TreeFARMS also results in enumerating sparse trees. Furthermore, it uses "GOSDT lookahead bounds" in actual that are not directly described in this paper.

---

> ### Author Response · Authors · 2022-08-02
> **Response to reviewer 4Gdc**
>
> Thank you so much for your review! We really appreciate it! See below for our response to your questions.
>
> > Q: Undefined procedures
>
> These procedures do exactly what is described in the comments. For example, gosdt.get\_graph returns the graph $G$ from GOSDT.
> We will add a complete description of those algorithms in the appendix.
>
> > Q. why the paper title includes "sparse" although Algorithm 1 and 2 have no sparsity-inducing parts apparently.
> If I understand it, the model search space for complete enumeration comes from GOSDT targeting sparse decision trees, and thus, TreeFARMS also results in enumerating sparse trees.
>
> The Rashomon set is defined in terms of proximity to an optimal tree. Like all algorithms that find optimal trees, GOSDT, and by extension TreeFARMS, uses an objective with a sparsity term to produce interpretable models that generalize well.
> TreeFARMS imposes a per-leaf penalty ($\lambda$ in the objective function), to favor trees with fewer leaves to those with more leaves. Thus, the calls to $obj$ in lines 11 and 12 of Algorithm 2, including this $\lambda$ term, thereby induce sparsity.
>
> > Furthermore, it uses "GOSDT lookahead bounds" in actual that are not directly described in this paper.
>
> Thank you for asking about this! The GOSDT lookahead bound reduces the search space, by eliminating portions of the space guaranteed not to contain the *optimal* tree (Theorem B.2 in [1]). In TreeFARMS, we must search a larger space to ensure that we find all trees in the Rashomon set. This requires a new lookahead bound (and implementation), that eliminates *only those trees guaranteed not to be in the Rashomon set*. We will include this explanation in the paper.
>
> > Q: Because the title includes "sparse", and thus could you explain why enumeration is limited to "sparse", and what does the paper mean by the word "sparse", or explicitly (but concisely?) explain the dependency inherited from GOSDT?
>
> As mentioned above, we use a per-leaf penalty in the objective to induce sparsity. In the TreeFARMS setting, sparsity is important for three reasons: 1) It ensures that the models we produce are interpretable, 2) it ensures that the models generalize well, and 3) it makes the task of enumeration feasible.
> As the number of trees grows exponentially with depth, enumeration of non-sparse trees is impractical.
> We will clarify that while GOSDT is designed to *eliminate almost all of* the search space (to produce one tree), TreeFARMS is designed to find and *store* the whole Rashomon set.
> We modified GOSDT to produce a dependency graph that contains the entire Rashomon set, rather than to simply guarantee that it contains the optimal tree. TreeFARMS then goes on to extract trees from the space represented by the (larger) dependency graph. We extract the trees using Sec 4.2 and store the trees using the new Model Set representation.
>
> > Q: Could you add the table of contents for the supplementary document?
>
> Definitely. We will add the table of contents for the supplementary document.
>
> > Q: Is the "dependency graph" actually a tree, a DAG, or a cyclic graph? I'm not sure about the connection, but I felt some abstract similarity to topics such as decision diagram (DD) representation of logical functions and traversing an enumeration tree, the search space of gSpan algorithm for all subgraph patterns, with bounds as in https://doi.org/10.1109/TPAMI.2016.2567399 for example.
>
> The dependency graph is a DAG. DAGs are common data structures for storing subproblems in dynamic programming. Figure 10 in Lin et al., 2020 shows the graph representation of the dependency graph for GOSDT. (DD is different as it starts with a single tree and enumerates only subtrees of it - the search space is limited to the given tree. However, our method has a larger search space, because it can extend to any sparse tree over all features. Note that decision trees can always be represented as decision diagrams as they are logical functions.)
> Unlike gSpan, our algorithm is not considering frequency of subgraphs - that's not our goal here.
>
>
> Thank you so much once again!

---

> > ### Comment · Reviewer_4Gdc · 2022-08-05
> > **Thanks!**
> >
> > Thanks for the clarification. This is just an acknowledgment that I've read the comment. The author's responses answered all unclear points I'd like to know, in particular, the per-leaf penalty, lambda term, corresponds to the sparsity-inducing factor and 'sparse' means less leaves for ensuring interpretability. Also, the fact that the dependency graph is a DAG ensured my view on the algorithm efficiency for why the presented algorithms made the complete enumeration of the Rashomon set possible. Also, I'm happy to see additional explanations on something like gosdt.get_graph and "GOSDT lookahead bound" in the revised paper.

---

### Official Review · Reviewer_N1V8 · 2022-07-12

**Rating:** 7
**Confidence:** 4
**Soundness:** 3 good
**Presentation:** 3 good
**Contribution:** 3 good

**Summary:**

The paper present a method called TreeFARMS, which is a combination of analytical bounds and a specialized representation that produces enumerations of the Rashomon set for sparse decision trees. This methodology provides insight into the size of the Rashomon set, how variable importance changes over the set, predictive consistency across rashomon sets, and rashomon sets for other losses.

**Questions:**

What was the motivation for using the model set representation?

**Limitations:**

The authors discuss the paper's limitations. They specifically clarify that the models should not be interpreted as causal and also focuses on sparse binary decisions trees as the model class.

**Strengths And Weaknesses:**

The paper presents bound to reduce the search space for Rashomon set construction. The basic lower bound and the rashomon equivalent points bound. They implement the algorithm using dynamic programming to find the optimal sparse decision trees. The work builds on existing literature focused on Rashomon sets and multiplicity to open an important discussion on variable importance and distributional shifts.

---

> ### Author Response · Authors · 2022-08-02
> **Response to reviewer N1V8**
>
> Thank you so much for your review! We really appreciate it! See below for our response to your questions.
>
> > Q: What was the motivation for using the model set representation?
>
>  The model set representation helps when the Rashomon sets are BIG. We actually initially implemented a trie-based representation that explicitly represented every tree. However, that structure limited our scalability. The Model Set representation was able to reduce memory consumption and runtime significantly and therefore scale to much larger Rashomon sets. For example, on the COMPAS dataset, with Rashomon multipler ($\epsilon$) of 0.15 and regularization of 0.01, the extraction time for the Rashomon set is less than 10 seconds using Model Sets; without Model Sets, extraction takes $10^4$ sec.
> We will include this discussion and comparisons in the final paper. Thank you so much once again for your review!

---

### Official Review · Reviewer_e6ES · 2022-07-12

**Rating:** 9
**Confidence:** 3
**Soundness:** 3 good
**Presentation:** 4 excellent
**Contribution:** 4 excellent

**Summary:**

The paper studies the problem of enumerating the Rashomon set for sparse decision trees.  The Rashomon set is the set of all almost-optimal models, and the paper provides the first result in enumerating it for highly nonlinear discrete models. The paper proposes an algorithm, TreeFARMS, to enumerate the Rashomon set for sparse decision trees, which stores the Rashomon set in a specialized and memory efficient data structure. The experimental evaluation, which considers several datasets from different domains, shows that the ability of enumerating the Rashomon set empowers the user with the ability to answer novel questions, such as: how does the variable importance change among trees within the Rashomon set? Do all almost-optimal models provide the same prediction for the same instance? How does the Rashomon set change if a subset of the data is removed?

**Questions:**

No questions

**Limitations:**

 There is no discussion, but the introduction provides some motivation for why the paper may have a positive impact.

**Strengths And Weaknesses:**

Strengths:
- overall, the paper provides a significant, high-quality, and original contribution
- the problem of enumerating the Rashomon set is very interesting in general, and very challenging for sparse decision trees
- the paper is clearly written and provides a clear motivation for the problem and a nice summary of the main contributions
- the methodological and algorithm contributions are sound
- the  experimental evaluation is clearly presented and comprehensive. It clearly shows how the proposed tool can be used to study the space of well-performing sparse decision trees

No weaknesses

---

> ### Author Response · Authors · 2022-08-02
> **Response to reviewer e6ES**
>
> Thank you so much for your review! We really appreciate it!

---

### Meta-Review · Area_Chair_Kra2 · 2022-08-22

**Recommendation:** Accept
**Confidence:** Certain

**Metareview:**

This paper presents an algorithm to completely enumerate the Rashomon set of sparse decision trees using  a branch-and-bound search over a set of hierarchical maps. The reviewers were in agreement that this paper is an important contribution to the field. The method is particularly useful for identifying near-optimal decision trees and variable importance scoring.

The problem of the Rashomon effect, where many equally good models could explain the data well, is a longstanding concern across nearly all machine learning applications. The Rashomon set is the set of such nearly-optimal models. While researchers would have liked to explore the Rashomon set, computational feasibility has limited such endeavors. This paper enables a shift in focus from a single optimal model to set of near-optimal models for a large and widely used class of models. Therefore, it can be expected that this work will have groundbreaking impact on multiple areas of machine learning.

**Award:**

Yes

---

### Decision · Program_Chairs · 2022-09-14

Accept